# Sentinel Lymph Node Biopsy vs. Observation in Thin Melanoma: A Multicenter Propensity Score Matching Study

**DOI:** 10.3390/jcm10245878

**Published:** 2021-12-15

**Authors:** Antonio Tejera-Vaquerizo, Aram Boada, Simone Ribero, Susana Puig, Sabela Paradela, David Moreno-Ramírez, Javier Cañueto, Blanca de Unamuno-Bustos, Ana Brinca, Miguel A. Descalzo-Gallego, Simona Osella-Abate, Paola Cassoni, Sebastian Podlipnik, Cristina Carrera, Sergi Vidal-Sicart, Ramón Pigem, Agustí Toll, Ramón Rull, Llucìa Alos, Celia Requena, Isidro Bolumar, Víctor Traves, Ángel Pla, Almudena Fernández-Orland, Ane Jaka, María Teresa Fernández-Figueras, Nina Anika Richarz, Ricardo Vieira, Rafael Botella-Estrada, Concepción Román-Curto, Lara Ferrándiz-Pulido, Nicolás Iglesias-Pena, Carlos Ferrándiz, Josep Malvehy, Pietro Quaglino, Eduardo Nagore

**Affiliations:** 1Dermatology Department, Instituto Dermatológico GlobalDerm, 14700 Palma del Río, Spain; 2Cutaneous Oncology Unit, Hospital San Juan de Dios, 14012 Córdoba, Spain; 3Dermatology Department, Hospital Universitari Germans Trials i Pujol, Institut d’Investigació Germans Trias i Pujol, Universitat Autònoma de Barcelona, 08916 Badalona, Spain; aramboada@gmail.com (A.B.); iratzia@hotmail.com (A.J.); richarznina@gmail.com (N.A.R.); cferrandizf@gmail.com (C.F.); 4Medical Sciences Department, Section of Dermatology, University of Turin, 10124 Turin, Italy; simone.ribero@unito.it (S.R.); pietro.quaglino@unito.it (P.Q.); 5Melanoma Unit, Dermatology Department, Hospital Clinic, Universitat de Barcelona, Institut d’Investigacions Biomèdiques August Pi i Sunyer (IDIBAPS), 08036 Barcelona, Spain; spuig@clinic.cat (S.P.); spodlipnik@gmail.com (S.P.); ccarrera@clinic.cat (C.C.); ramon.pigem@gmail.com (R.P.); atoll@clinic.cat (A.T.); jmalvehy@gmail.com (J.M.); 6Biomedical Research Networking Center on Rare Diseases (CIBERER), ISCIII, 28029 Barcelona, Spain; 7Departamento de Dermatología, Hospital Universitario de la Coruña, 15006 La Coruña, Spain; sparmor78@gmail.com (S.P.); niglep@gmail.com (N.I.-P.); 8Melanoma Unit, Medical-&-Surgical Dermatology Department, Hospital Universitario Virgen Macarena, 41009 Sevilla, Spain; david.moreno.ramirez.sspa@juntadeandalucia.es (D.M.-R.); almudenaorland@yahoo.com (A.F.-O.); lferrandiz@e-derma.org (L.F.-P.); 9Dermatology Department, Complejo Asistencial Universitario de Salamanca, 37007 Salamanca, Spain; javier.canueto@gmail.com (J.C.); cromancurto@gmail.com (C.R.-C.); 10Dermatology Department, Hospital Universitario La Fe, 46126 Valencia, Spain; blancaunamuno@yahoo.es (B.d.U.-B.); rbotellaes@gmail.com (R.B.-E.); 11Departament of Dermatology, University Hospital of Coimbra, 3000-075 Coimbra, Portugal; anabrinca@gmail.com (A.B.); ricardo.jdc.vieira@gmail.com (R.V.); 12Unidad de Investigación, Fundación Piel Sana, Academia Española de Dermatología, 28008 Madrid, Spain; miguelangel.descalzo@aedv.es; 13Medical Sciences Department, Section of Surgical Pathology, University of Turin, 10124 Turin, Italy; simona.osellaabate@unito.it (S.O.-A.); paola.cassoni@unito.it (P.C.); 14Nuclear Medicine Department, Hospital Clinic Barcelona, Universitat de Barcelona, Institut d’investigacions Biomèdiques August Pi i Sunyer (IDIBAPS), 08036 Barcelona, Spain; svidalsicart@gmail.com; 15Surgery Department, Hospital Clinic, Universidad de Barcelona, 08036 Barcelona, Spain; RRULL@clinic.cat; 16Pathology Department, Hospital Clinic, Universidad de Barcelona, 08036 Barcelona, Spain; lalos@clinic.cat; 17Dermatology Department, Instituto Valenciano de Oncología, 46009 Valencia, Spain; celiareq@hotmail.com (C.R.); eduardo_nagore@ono.com (E.N.); 18Surgery Department, Instituto Valenciano de Oncología, 46009 Valencia, Spain; ibolumar@yahoo.es; 19Pathology Department, Instituto Valenciano de Oncología, 46009 Valencia, Spain; victortraves@telefonica.net; 20Otorhinolaringology Department, Instituto Valenciano de Oncología, 46009 Valencia, Spain; apmocholi@gmail.com; 21Pathology Department, Hospital Universitari Germans Trial i Pujol, 08916 Badalona, Spain; maiteffig@gmail.com; 22Instituto de Investigación Biomédica de Salamanca, Complejo Asistencial Universitario de Salamanca, 37007 Salamanca, Spain

**Keywords:** melanoma, sentinel lymph node biopsy, survival

## Abstract

The therapeutic value of sentinel lymph node biopsy (SLNB) in thin melanoma remains controversial. The aim of this study is to determine the role of SLNB in the survival of thin melanomas (≤1 mm). A multicenter retrospective observational study was designed. A propensity score matching was performed to compare patients who underwent SLNB vs. observation. A multivariate Cox regression was used. A total of 1438 patients were matched by propensity score. There were no significant differences in melanoma-specific survival (MSS) between the SLNB and observation groups. Predictors of MSS in the multivariate model were age, tumor thickness, ulceration, and interferon treatment. Results were similar for disease-free survival and overall survival. The 5- and 10-year MSS rates for SLN-negative and -positive patients were 98.5% vs. 77.3% (*p* < 0.001) and 97.3% vs. 68.7% (*p* < 0.001), respectively. SLNB does not improve MSS in patients with thin melanoma. It also had no impact on DSF or OS. However, a considerable difference in MSS, DFS, and OS between SLN-positive and -negative patients exists, confirming its value as a prognostic procedure and therefore we recommend discussing the option of SLNB with patients.

## 1. Introduction

Sentinel lymph node biopsy (SLNB) is a commonly used procedure in the management of cutaneous melanoma [1]. Although the Multicenter Selective Lymphadenectomy Trial I (MSTL-I) showed that SLNB does not improve disease-specific survival in melanoma (MSS) [2], it did not include tumors with a Breslow thickness < 1.2 mm in the analysis. The therapeutic effect of SLNB in thin melanoma thus remains to be determined. This is important, particularly in our setting, where tumors with a Breslow thickness < 1 mm are the most common diagnosed melanomas [3]. In addition, SLNB is recommended for patients with stage T1b melanoma and stage T1a melanoma if there are other high-risk factors, such as a mitotic rate > 2 mitoses/mm^2^ [4], lymphovascular invasion, and young age [5].

The main aim of this study was to determine whether SLNB improves MSS in patients with thin tumors. Secondary objectives were to compare disease-free survival (DFS) and overall survival (OS) between patients who undergo SLNB and those who undergo observation and to examine the effect of SLN positivity on survival. 

## 2. Materials and Methods

### 2.1. Study Population

We designed a multicenter observational study following the Strengthening the Reporting of Observational Studies in Epidemiology guidelines [6]. 

Patients were selected from the databases of nine hospitals that form part of the Sentinel Lymph Node Study Group in Melanoma (Sentimel). Seven of the hospitals are in Spain: Instituto Valenciano de Oncología in Valencia, Hospital Universitario de Salamanca in Salamanca, Hospital La Fe in Valencia, Hospital Universitario Virgen Macarena in Seville, Hospital de la Coruña in A Coruña, Hospital Germans Trias i Pujol in Badalona, and Hospital Clínic in Barcelona. The other two hospitals are in Portugal (Centro Hospitalar e Universitário de Coimbra) and Italy (University Hospital “Città della Salute e della Scienza di Torino”). 

We included all patients aged ≥18 years who were registered in the hospital databases up to 31 December 2017 with a diagnosis of thin melanoma (Breslow thickness ≤ 1 mm) and no evidence of metastasis at diagnosis. 1 January 1998 was chosen as the start date for inclusion, as this is when most hospitals started to use SLNB in the management of melanoma [7]. SLNB is performed using a similar procedure at all the hospitals with any combination of vital blue dye, radioactive tracer, and preoperational lymphography (+/− preoperative PET-CT/CT) for SLN mapping. Thin primary melanomas are excised with a 1-cm margin, as recommended by clinical practice guidelines. The procedure for pathologic SLN examination has been described previously [8]. Hospital Clínic in Barcelona has been using the Minitub protocol (EORTC 1208: Minitub registration study) since 2011. The study was approved by the lead ethics committee, located at Hospital Universitario Reina Sofía in Cordoba (reference 3569). 

### 2.2. Study Groups and Outcome Variables 

The patients were divided into two groups: an SLNB group and an observation-only group. Patients in the SLNB group were further classified as SLN-positive or SLN-negative. 

The outcome variables were DFS, MSS, and OS. Survival was defined as time in months from excision of the primary tumor to first recurrence (DFS), death due to melanoma (MSS), or death due to any cause (OS). Recurrence was classified as local recurrence or satellite, regional lymph node recurrence, or distant metastasis. In patients with multiple simultaneous recurrences, the most advanced type of recurrence was considered. 

### 2.3. Propensity Score Matching 

Propensity score matching is a relatively new statistical technique that controls for selection biases in non-randomized studies comparing two interventions or treatments [9]. It consists of matching patients according to their likelihood of being assigned to one group or another, in our case: SLNB or observation. The first step was to perform logistic regression with SLNB as the dependent variable and all the other variables as independent variables. The independent variables were chosen because of their potential prognostic value in melanoma [10] and comprised Breslow thickness [11], ulceration [11,12], regression [13,14], Clark level, microscopic satellitosis [15], mitotic rate [16], vascular invasion [16], tumor infiltrating lymphocytes [17], histologic subtype, age, sex, anatomic location [18], hospital, year, and treatment with interferon [19]. Histologic subtypes were “superficial spreading melanoma”, “nodular melanoma” and “other” histological subtypes. For the convergence of the models it was mandatory to reunify the rest of the histologic subgroups (lentigo maligna, acral lentiginous melanoma, …) into a single simple group (other).

### 2.4. Statistical Analysis 

Between-group comparisons were made using the Mann–Whitney U test and the *t* test for qualitative and quantitative variables respectively. Breslow thickness and age were log-transformed to avoid skewed distribution. Separate models were built for DFS, MSS, and OS. Survival times were calculated from excision of the primary tumor to the event in question. Cases with no events up to the date of the last follow-up were treated as censored data. Survival curves estimated using the Kaplan–Meier method were compared using the log-rank test to compare survival between patients in the SLNB and observation-only groups. The same method was used to compare SLN-positive and SLN-negative patients. Univariate Cox regression was used to assess the effect of each variable on survival according to the performance of SLNB or not. A multivariate model was built to analyze the impact on survival of all variables with a significance level of *p* < 0.2 in the univariate analysis. 

### 2.5. Missing Data

Assuming that missing data were missing at random, we generated 20 complete datasets using multivariate imputation by chained equations (mi impute chained procedure in Stata). The procedure included all variables that were to be subsequently analyzed in addition to any variables that could help explain the missing data. Each of 20 imputed datasets was analyzed using Cox regression to fit the model of interest to the outcome variables (DFS, MSS, and OS). Finally, the results of the complete datasets were combined into a single set of estimates using Rubin rules [20]. All analyses were performed in STATA v.14.1 (Stata Corp. 2015. Stata Statistical Software: Release 14).

## 3. Results

### 3.1. Study Population Characteristics 

We included 5049 patients with thin localized melanoma (≤1 mm) at diagnosis; 1083 had undergone SLNB and 3966 observation only (Figure 1). In total, 1438 patients were matched by propensity scores. 

Before matching, patients in the observation group were more likely to be women (58% vs. 53%, *p* < 0.001) and to have melanoma of the head and neck (14% vs. 7%, *p* < 0.001) and less likely to have ulceration (1% vs. 10%, *p* < 0.001), regression (26% vs. 45%, *p* < 0.001), and a Clark level IV (6% vs. 26%, *p* < 0.001). They also had lower mitotic rates. There were no significant differences between the groups after matching (Table 1).

### 3.2. Survival Rates 

Median follow-up was 61 months. During this time, there were 82 recurrences (5.7%), 46 melanoma-specific deaths (3.2%), and 74 deaths due to another cause (5.1%); 8.3% of patients in the SLNB group and 10.3% of those in the observation group were lost to follow-up. 

There were no significant differences in MSS between the SLNB and observation groups. The respective 5- and 10-year survival rates were 97.4% vs. 97.1 % and 95.3% vs. 95.6%. The corresponding 5- and 10-year rates for DFS were 95.3% vs. 94.3% and 90.8% vs. 91.8%. The differences for 5-year and 10-year OS were also non-significant (Figure 2). 

When all other variables were controlled for in the multivariate analysis, SLNB was not a significant predictor of either MSS or DFS. It was, however, an independent predictor of OS (adjusted hazard ratio, 0.61; 95% CI, 0.37–1; *p* = 0.05).

#### 3.2.1. Melanoma-Specific Survival

The predictors of MSS in the multivariate model were age, tumor thickness, ulceration, and interferon treatment (Table 2). 

#### 3.2.2. Disease-Free Survival

The independent predictors of DFS were age, histologic subtype other than superficial spreading melanoma and nodular melanoma, ulceration, Clark level, mitotic rate, and interferon treatment (Table 3). 

#### 3.2.3. Overall Survival

The independent predictors of OS, in addition to SLNB, were sex, age, ulceration, Clark level, and interferon treatment (Table 4). 

### 3.3. Prognostic Significance of SLNB

Forty-two patients in the SLNB group (5.8%) were SLN-positive, but seven false negatives were detected during follow-up. The overall false negative rate was 14.3%, which was calculated by dividing the number of false negatives by the sum of positive cases and false negatives according to the method described by van Akkooi et al. [21].

The 5- and 10-year MSS rates for SLN-negative and -positive patients were 98.5% vs. 77.3% (*p* < 0.001) and 97.3% vs. 68.7% (*p* < 0.001), respectively. The corresponding rates for the other survival categories were 96.6% vs. 60.9% (*p* < 0.001) and 94.6% vs 48.9% (*p* < 0.001) for DFS and 97.3% vs. 78.9% (*p* < 0.001) and 95.5% vs. 66.6% (*p* < 0.001) for OS (Figure 3). 

## 4. Discussion

The main conclusion of this study is that SLNB does not improve MSS in patients with thin melanoma. It also had no impact on DSF or OS. The conclusion for MSS is the same as that reached in the MSLT-I [2] but for melanomas with a Breslow thickness < 1 mm. 

The theoretical basis for the introduction of SLNB in the treatment of cutaneous melanoma in the 1990s was that the regional lymph nodes act as an incubator for subsequent distant spread [22]. Our focus on thin melanomas is justified as these tumors have a different pattern of spread. Compared with thicker melanomas, they have a greater propensity for locoregional metastasis and are less likely to spread to distant sites [21]. Our results, however, indicate the presence of synchronous regional and distant metastasis in thin melanoma, which would explain the absence of a significant survival benefit for SLNB in this setting [23,24,25]. 

Very few observational studies have analyzed the impact of SLNB on survival in patients with thin melanoma. Using data from the Surveillance, Epidemiology, and End Results (SEER) program, Sperry et al. [26] found no difference in MSS in 1104 propensity score-matched patients with thin melanoma who had undergone SLNB or nodal observation. More recently, Murtha et al. [27], using the same database, reported a significant difference in OS but not MSS over a median follow-up period of 16 months for a population of 3439 patients with melanoma with a thickness of 0.75 to <1 mm. Finally, in another propensity score matching study using data from the US National Cancer Data Base, Sinnamon et al. [28] found no differences in OS between 4262 pairs of melanoma patients with a Breslow thickness of 0.5 to 0.7 mm. They did, however, find a difference for OS among patients with tumors measuring 0.8 to 1.0 mm. One limitation of their study, however, was that the database does not contain information on MSS. 

Our analysis of SLN-positive and -negative patients show worse survival rates than in similar studies [29,30], probably because of differences in patient selection criteria. These differences in survival could justify the use of new adjuvant therapies and should be discussed with patients [5]. 

The main limitation of this study is its retrospective design. The groups may not have been properly balanced as not all potential confounders were considered (e.g., comorbidities and performance status). Our study may also be underpowered, as it has been calculated that 6500 patients would be needed to detect a protective effect for SLNB using a similar design to the MSLT-1, based on a power of 90%, a follow-up period of 5 years, and an estimated hazard ratio of 0.8 for SLNB [26]. Furthermore, we did not specifically analyze time to recurrence at regional lymph nodes; the only expected benefit of the SLNB according to previous studies focused on thicker tumors.

Ulceration and thickness remain as independent prognostic factors associated with MSS survival in thin melanomas. Age also remains as an independent prognostic factor of MSS. It has been evidenced that patients at extreme age have a distinct natural history [31,32]. These data are congruent with the AJCC as ulceration is established as a variable that increases the staging according to a certain thickness while melanoma thicknesses close to 1 mm are already considered another stage [7]. It remains to be seen whether advanced age may contribute in the future to defining these melanomas with a worse prognosis.

The fact that interferon treatment is associated to worse MSS (HR 7.29 *p* < 0.001) should be considered as subsidiary of positivity of the SLNB, because the result of the procedure was not included in the analysis and interferon treatment was only indicated in the cases of lymph node positivity in thin melanomas as indicated in the active guide lines during the period of the study.

## 5. Conclusions

SLNB is currently used for staging purposes in thin melanoma. Our study of a large cohort of patients with thin melanoma did not show that SLNB modifies survival in this setting. We did, however, observe a considerable difference in MSS, DFS, and OS between SLN-positive and -negative patients and therefore recommend discussing the option of SLNB with patients.

## Figures and Tables

**Figure 1 jcm-10-05878-f001:**
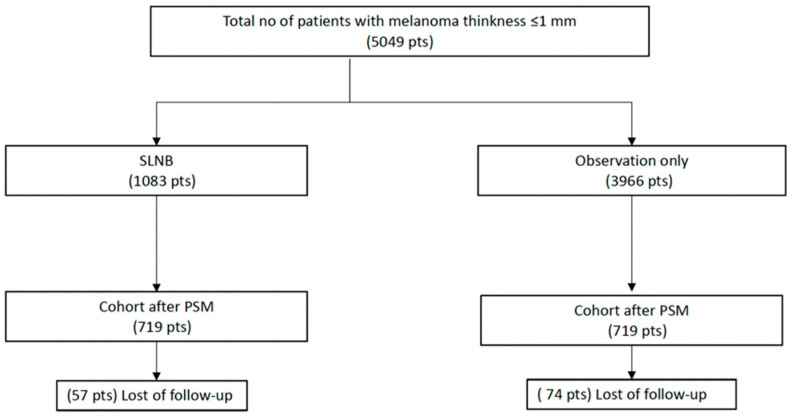
Flow chart of the study population. PSM denotes propensity score matching; SLNB, sentinel lymph node biopsy.

**Figure 2 jcm-10-05878-f002:**
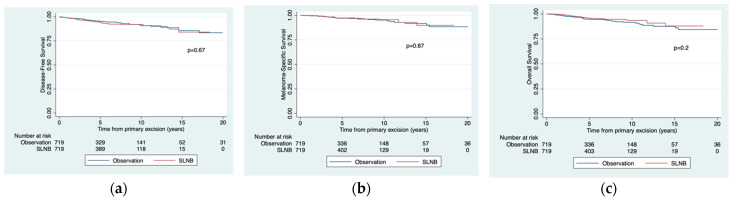
Estimated disease-free survival (**a**), melanoma-specific survival (**b**), and overall survival (**c**) according to study group. Survival curves calculated using the Kaplan–Meier method according to study group in the propensity score-matched sample (n = 1438). SLNB denotes sentinel lymph node biopsy.

**Figure 3 jcm-10-05878-f003:**
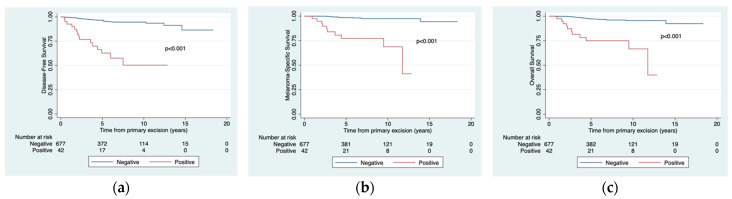
Estimated disease-free survival (**a**), melanoma-specific survival (**b**), and overall survival (**c**) according to sentinel lymph node status.

**Table 1 jcm-10-05878-t001:** Characteristics of patients with thin cutaneous melanoma (<1 mm) according to study group (SLNB vs observation) before and after propensity score matching.

Characteristics	Before Propensity Score Matching			After Propensity Score Matching		
	OBSERVATION	SLNB	*p*-Value	OBSERVATION	SLNB	*p*-Value
	*n* = 3966	*n* = 1083		*n* = 719	*n* = 719	
**Year**	***N* (%)**	***N* (%)**	<0.001	***N* (%)**	***N* (%)**	0.675
≤2000	1295 (33)	97 (9)		103 (14)	88 (12)	
2001–2006	827 (21)	294 (27)		186 (26)	197 (27)	
2007–2011	986 (25)	334 (31)		227 (32)	227 (32)	
2012–2017	858 (22)	358 (33)		203 (28)	207 (29)	
**Hospital**						
Salamanca	108 (3)	21 (2)	<0.001	12 (2)	14 (2)	0.544
Valencia IVO	440 (11)	218 (20)		119 (17)	132 (18)	
Turin	1494 (38)	296 (27)		187 (26)	204 (28)	
Barcelona	1017 (26)	202 (19)		163 (23)	160 (22)	
Badalona	361 (9)	194 (18)		133 (18)	105 (15)	
Coimbra	65 (2)	14 (1)		14 (2)	12 (2)	
A Coruña	202 (5)	74 (7)		51 (7)	43 (6)	
Sevilla	203 (5)	40 (4)		23 (3)	31 (4)	
Valencia La Fe	76 (2)	24 (2)		17 (2)	18 (3)	
**Sex**			0.001			0.459
Male	1646 (42)	509 (47)		324 (45)	338 (47)	
Female	2320 (58)	573 (53)		395 (55)	381 (53)	
**Mean age (sd), y**	52.4 (16.4)	51.9 (14.7)	0.3372	52.9 (16.7)	52.5 (14.9)	0.6422
**Tumor location**			<0.001			0.266
Head/neck	528 (14)	80 (7)		86 (12)	67 (9)	
Trunk	1579 (41)	473 (44)		300 (42)	307 (43)	
Extremities (upper and lower)	1725 (45)	518 (48)		333 (46)	345 (48)	
**Log tumor thickness. median (p25–p75)**	−0.7 (−0.9–−0.4)	−0.2 (−0.5–−0.1)	<0.001	−0.4 (−0.6–−0.2)	−0.7 (−0.3–−0.1)	0.063
**Histologic subtype**			<0.001			0.623
Superficial spreading melanoma	3293 (84)	871 (81)		578 (80)	592 (82)	
Nodular melanoma	45 (1)	54 (5)		23 (3)	22 (3)	
Other	588 (15)	145 (14)		118 (16)	105 (15)	
**Ulceration**			<0.001			0.368
No	3467 (99)	890 (90)		673 (94)	681 (95)	
Yes	50 (1)	100 (10)		46 (6)	38 (5)	
**Regression**			<0.001			0.872
No	2422 (74)	494 (55)		425 (59)	428 (60)	
Yes	845 (26)	399 (45)		294 (41)	291 (40)	
**Microscopic satellite**			0.1265			1.000
No	1529 (100)	584 (99)		710 (99)	711 (99)	
Yes	4 (0)	5 (1)		9 (1)	8 (1)	
**Tumor infiltrating lymphocytes**			0.5505			0.473
No	272 (27)	67 (27)		192 (27)	175 (24)	
Non-brisk	580 (57)	134 (54)		377 (52)	399 (55)	
Brisk	169 (17)	48 (19)		150 (21)	145 (20)	
**Vascular invasion**			0.744			0.803
No	1655 (100)	608 (99)		712 (99)	710 (99)	
Yes	8 (0)	4 (1)		7 (1)	9 (1)	
**Interferon treatment**			<0.001			0.358
No	2361 (99)	738 (95)		700 (97)	694 (97)	
Yes	20 (1)	37 (5)		19 (3)	25 (3)	
**Clark level**			<0.001			0.838
I-III	2897 (94)	755 (74)		588 (82)	585 (81)	
IV	184 (6)	263 (26)		131 (18)	134 (19)	
**Mitotic rate (mitoses/mm^2^)**			<0.001			0.991
0	1491 (82)	248 (34)		341 (47)	339 (47)	
1	225 (12)	282 (39)		247 (34)	252 (35)	
2	55 (3)	116 (16)		84 (12)	81 (11)	
≥3	49 (3)	82 (11)		47 (7)	47 (7)	


Log, logarithm; SLNB, sentinel lymph node biopsy.

**Table 2 jcm-10-05878-t002:** Univariate and multivariate analysis of predictors of melanoma-specific survival in patients included in the study (*n* = 1438).

	Crude Univariate Analysis				Adjusted Multivariate Analysis				
	HR	95% CI LL	95% CI UL	*p*-Value		HR	95% CI LL	95% CI UL	*p*-Value
**SLNB**					**SLNB**				
No	Ref	-	-	-	No	Ref	-	-	-
Yes	0.96	0.53	1.73	0.884	Yes	0.84	0.45	1.56	0.575
**Year**									
≤2000	Ref	-	-	-					
2001–2006	0.73	0.35	1.54	0.410					
2007–2011	0.98	0.42	2.27	0.958					
2012–2017	0.67	0.18	2.54	0.558					
**Hospital**									
Salamanca	Ref	-	-	-					
Valencia IVO	0.54	0.07	4.29	0.562					
Turin	0.48	0.06	3.74	0.487					
Barcelona	0.74	0.10	5.71	0.772					
Badalona	0.39	0.05	3.22	0.379					
Coimbra	NA								
A Coruña	0.25	0.02	4.01	0.328					
Sevilla	NA								
Valencia La Fe	NA								
**Sex**									
Male	Ref	-	-	-					
Female	0.57	0.31	1.02	0.057					
**Age**	1.02	1.00	1.04	0.076	**Age**	1.03	1.01	1.05	0.011
**Log age**	1.98	0.73	5.34	0.18					
**Tumor location**									
Head/neck	Ref	-	-	-					
Trunk	1.39	0.48	4.02	0.541					
Extremities (upper and lower)	0.85	0.29	2.53	0.771					
**Tumor thickness**	13.71	2.68	69.96	0.002					
**Log tumor thickness**	4.76	1.56	14.51	0.006	**Log tumor thickness**	3.82	1.23	11.81	0.020
**Histologic subtype**									
Superficial spreading melanoma	Ref	-	-	-					
Nodular melanoma	4.10	1.59	10.62	0.004					
Other	1.94	0.92	4.08	0.082					
**Ulceration**					**Ulceration**				
No	Ref	-	-	-	No	Ref	-	-	-
Yes	3.23	1.43	7.28	0.005	Yes	2.66	1.11	6.38	0.028
**Regression**									
No	Ref	-	-	-					
Yes	0.97	0.50	1.89	0.922					
**Microscopic satellite**									
No	Ref	-	-	-					
Yes	5.41	0.72	40.51	0.098					
**Tumor infiltrating lymphocytes**									
1	Ref	-	-	-					
2	0.90	0.41	1.98	0.787					
3	0.62	0.14	2.67	0.517					
**Vascular invasion**									
No	Ref	-	-	-					
Yes	NA								
**Interferon**					**Interferon**				
No	Ref	-	-	-	No	Ref	-	-	-
Yes	7.70	3.45	17.19	0.000	Yes	7.29	2.94	18.06	0.000
**Clark level**									
I-II-III	Ref	-	-	-					
IV	1.92	0.93	3.95	0.076					
**Mitotic rate**									
0	Ref	-	-	-					
1	1.67	0.66	4.20	0.276					
2	2.28	0.63	8.20	0.204					
≥3	4.16	1.34	12.89	0.014					

HR, hazard ratio; LL, lower limit; Log, logarithm; SLNB, sentinel lymph node biopsy; UL, upper limit.

**Table 3 jcm-10-05878-t003:** Univariate and multivariate analysis of predictors of disease-free survival in patients included in the study (*n* = 1438).

	Crude Univariate Analysis					Adjusted Multivariate Analysis			
	HR	95% CI LL	95% CI UL	*p*-Value		HR	95% CI LL	95% CI UL	*p*-Value
**SLNB**					**SLNB**				
No	Ref	-	-	-	No	Ref	-	-	-
Yes	1.11	0.72	1.73	0.634	Yes	0.84	0.49	1.43	0.509
**Year**									
≤2000	Ref	-	-	-					
2001–2006	0.67	0.39	1.15	0.142					
2007–2011	0.62	0.34	1.14	0.124					
2012–2017	0.42	0.17	1.07	0.068					
**Hospital**									
Salamanca	Ref	-	-	-					
Valencia IVO	1.34	0.18	10.01	0.776					
Turin	1.25	0.17	9.21	0.829					
Barcelona	1.32	0.18	9.89	0.785					
Badalona	0.58	0.07	4.64	0.606					
Coimbra	NA								
A Coruña	0.50	0.05	5.53	0.573					
Sevilla	NA								
Valencia La Fe	0.95	0.06	15.16	0.970					
**Gender**									
Male	Ref	-	-	-					
Female	0.73	0.47	1.12	0.152					
**Age**	1.02	1.01	1.04	0.006	**Age**	1.03	1.01	1.04	0.003
**Log Age**	2.47	1.17	5.23	0.018					
**Localization**									
Head/Neck	Ref	-	-	-					
Trunk	0.95	0.44	2.07	0.907					
Extremities (upper and lower)	1.02	0.48	2.17	0.965					
**Tumor thickness**	7.32	2.33	23.06	0.001					
**Log tumor thickness**	3.03	1.43	6.40	0.004					
**Histologic subtype**					**Histologic subtype**				
Superficial spreading melanoma	Ref	-	-	-	Superficial spreading melanoma	Ref	-	-	-
Nodular melanoma	5.40	2.73	10.67	0.000	Nodular melanoma	1.58	0.56	4.47	0.389
Others	2.36	1.39	4.01	0.001	Others	2.51	1.36	4.63	0.003
**Ulceration**					**Ulceration**				
No	Ref	-	-	-	No	Ref	-	-	-
Yes	4.14	2.34	7.33	0.000	Yes	3.06	1.40	6.70	0.005
**Regression**									
No	Ref	-	-	-					
Yes	0.69	0.41	1.16	0.161					
**Microscopic satellite**									
No	Ref	-	-	-					
Yes	5.54	1.15	26.58	0.033					
**Tumor infiltrating lymphocytes**									
No	Ref	-	-	-					
Non-brisk	1.02	0.53	1.94	0.960					
Brisk	0.91	0.28	2.94	0.873					
**Vascular invasion**									
No	Ref	-	-	-					
Yes	NA								
**Interferon**					**Interferon treatment**				
No	Ref	-	-	-	No	Ref	-	-	-
Yes	10.80	5.97	19.52	0.000	Yes	15.12	7.36	31.07	0.000
**Clark level**					**Clark level**				
I-II-III	Ref	-	-	-	I-II-III	Ref	-	-	-
IV	2.17	1.30	3.61	0.003	IV	2.38	1.35	4.18	0.003
**Mitotic rate**					**Mitotic rate**				
0	Ref	-	-	-	0	Ref	-	-	-
1	1.72	0.85	3.50	0.131	1	2.03	0.93	4.42	0.074
2	3.06	1.26	7.44	0.014	2	3.08	1.15	8.21	0.025
≥3	7.30	3.38	15.78	0.000	3 or more	7.66	3.02	19.45	0.000

HR, hazard ratio; LL, lower limit; Log, logarithm; SLNB, sentinel lymph node biopsy; UL, upper limit.

**Table 4 jcm-10-05878-t004:** Univariate and multivariate analysis of predictors of overall survival in patients included in the study (*n* = 1438).

	Crude Univariate Analysis					Adjusted Multivariate Analysis			
	HR	LL 95%CI	UL 95%CI	*p*-Value		HR	LL 95%CI	UL 95%CI	*p*-Value
**SLNB**					**SLNB**				
No	Ref	-	-	-	No	Ref	-	-	-
Yes	0.74	0.46	1.19	0.211	Yes	0.61	0.37	1.00	0.050
**Year**									
≤2000	Ref	-	-	-					
2001–2006	1.06	0.56	1.99	0.857					
2007–2011	1.15	0.56	2.36	0.710					
2012–2017	1.69	0.71	4.02	0.236					
**Hospital**									
Salamanca	Ref	-	-	-					
Valencia IVO	0.97	0.13	7.37	0.980					
Turin	0.47	0.06	3.62	0.468					
Barcelona	1.11	0.15	8.32	0.922					
Badalona	0.93	0.12	7.10	0.944					
Coimbra	3.29	0.20	53.37	0.402					
A Coruña	1.25	0.15	10.71	0.838					
Sevilla	0.66	0.06	7.34	0.738					
Valencia La Fe	NA								
**Sex**					**Sex**				
Male	Ref	-	-	-	Male	Ref	-	-	-
Female	0.41	0.25	0.67	0.000	Female	0.48	0.29	0.79	0.004
**Age**	1.05	1.04	1.07	0.000	**Age**	1.05	1.03	1.07	0.000
**Log age**	10.71	4.15	27.62	0.000					
**Tumor location**									
Head/neck	Ref	-	-	-					
Trunk	0.84	0.40	1.75	0.646					
Extremities (upper and lower)	0.68	0.33	1.41	0.302					
**Tumor thickness**	4.47	1.40	14.33	0.012					
**Log tumor thickness**	2.38	1.12	5.03	0.023					
**Histologic subtype**									
Superficial spreading melanoma	Ref	-	-	-					
Nodular melanoma	2.36	0.94	5.94	0.068					
Other	1.52	0.83	2.79	0.177					
**Ulceration**					**Ulceration**				
No	Ref	-	-	-	No	Ref	-	-	-
Yes	2.75	1.41	5.38	0.003	Yes	2.58	1.25	5.34	0.011
**Regression**									
No	Ref	-	-	-					
Yes	1.32	0.81	2.18	0.267					
**Microscopic satellite**									
No	Ref	-	-	-					
Yes	3.43	0.51	23.23	0.202					
**Tumor infiltrating lymphocytes**									
No	Ref	-	-	-					
Non-Brisk	0.73	0.38	1.41	0.343					
Brisk	0.49	0.15	1.68	0.254					
**Vascular invasion**									
No	Ref	-	-	-					
Yes	NA								
**Interferon treatment**					**Interferon treatment**				
No	Ref	-	-	-	No	Ref	-	-	-
Yes	4.28	2.02	9.05	0.000	Yes	5.69	2.43	13.31	0.000
**Clark level**					**Clark level**				
I-II-III	Ref	-	-	-	I-II-III	Ref	-	-	-
IV	2.00	1.15	3.45	0.014	IV	1.86	1.06	3.27	0.031
**Mitotic rate**									
0	Ref	-	-	-					
1	1.57	0.80	3.09	0.190					
2	1.91	0.75	4.89	0.176					
≥3	2.61	1.04	6.54	0.041					

HR, hazard ratio; LL, lower limit; Log, logarithm; SLNB, sentinel lymph node biopsy; UL, upper limit.

## Data Availability

The data presented in this study are available on request from the corresponding author.

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
