# Peer review of "Sentinel Lymph Node Biopsy vs. Observation in Thin Melanoma: A Multicenter Propensity Score Matching Study"

_jcm, 2021, doi:10.3390/jcm10245878_

Round 1

Reviewer 1 Report

Thanks for inviting me to review this large SLNB Study on patients with thin melanoma.

As the authors point out, that large numbers involved in this study provide an important database for our increasing knowledge on melanoma SLNB.

In the introduction, SLNB is described as widely accepted. 
the ongoing controversy on SLNB is apparent. There is a large body of scientific opinion and evidence that SLNB should not be considered accepted.

I would think a more accurate description in the introduction to be ‘commonly used’ - a term without dispute.
The authors incorrectly state that MSLT1 did not include patients with thin melanoma. That study did include patients with Breslow thickness melanoma less than 1.2 mm.  It is just that this data set is yet to be published. 

The retrospective and non randomised nature of this study is clearly the large limitation, as stated by the authors. 

The authors attempt to in part address this through a relatively novel statistical method, the propensity score matching. This method has in itself major drawbacks and is probably more a further distraction rather than assistance. 
The paper may be better served to simply identify outcomes based on treatment received, - SLNB or no SLNB. 

The manuscript could describe the fact that that certain patients might be preselected for SLNB given other features of their condition. The reader can then choose or not to factor this aspect. 

Given (the now expected post other research) no survival difference between SLNB and no SLNB, the need for propensity score matching is even less helpful.

in the abstract, “thin” melanoma is mentioned and not described. The definition of this being less than 1.2 mm Breslow thickness should be stated in abstract. 

Author Response

Dear editor,

We would like to thank the reviewers for his comments which have contributed to the improvement in our manuscript.

Reviewer 1

Thanks for inviting me to review this large SLNB Study on patients with thin melanoma.

As the authors point out, that large numbers involved in this study provide an important database for our increasing knowledge on melanoma SLNB.

In the introduction, SLNB is described as widely accepted. 
the ongoing controversy on SLNB is apparent. There is a large body of scientific opinion and evidence that SLNB should not be considered accepted.

I would think a more accurate description in the introduction to be ‘commonly used’ - a term without dispute.

Answer: We agree with referee in this appreciation. We have modified the manuscript as follows:  Sentinel lymph node biopsy (SLNB) is a commonly used procedure in the management of cutaneous melanoma[1].

The authors incorrectly state that MSLT1 did not include patients with thin melanoma. That study did include patients with Breslow thickness melanoma less than 1.2 mm.  It is just that this data set is yet to be published. 

Answer: The reviewer is correct when affirming that MSLT1 included patient with melanoma less than 1.2 mm (candidates for inclusion were patients who had localized cutaneous melanomas of Clark level III with a Breslow thickness of 1.00 mm or more or melanomas of Clark level IV or V with any Breslow thickness. Therefore, it did not include patients with thin melanoma (<1 mm) if the Clark invasion level did not reach IV.  Consequently, the vast majority of patients with thin melanoma would not have been eligible for the MSLT1. In any case, the data referring to the patients with thin melanoma included in MSLT1 have never been published.

We have modified the manuscript as follows: Although the Multicenter Selective Lymphadenectomy Trial I (MSTL-I) showed that SLNB does not improve disease-specific survival in melanoma (MSS) [2], it did not include in the analysis, tumors with a Breslow thickness <1.2 mm.

The retrospective and non randomised nature of this study is clearly the large limitation, as stated by the authors. 
The authors attempt to in part address this through a relatively novel statistical method, the propensity score matching. This method has in itself major drawbacks and is probably more a further distraction rather than assistance. 
The paper may be better served to simply identify outcomes based on treatment received, - SLNB or no SLNB. 
The manuscript could describe the fact that that certain patients might be preselected for SLNB given other features of their condition. The reader can then choose or not to factor this aspect. 
Given (the now expected post other research) no survival difference between SLNB and no SLNB, the need for propensity score matching is even less helpful.

Answer: In response to the reviewer in these paragraphs, we would like to say that due to the lack of a clinical trial aimed at analyzing the role of SLNB in thin melanoma we chose propensity score matching in an attempt to simulate randomization in a clinical trial setting. Obviously, this is not a perfect analysis, but in our opinion it is a valid statistical method (more accurate than direct comparison between groups) in accordance with the aim of the study. 

Moreover, the two groups before pairing were far from being comparable. Before matching, patients in the observation group were more likely to be women (58% vs. 53%, p<0.001) and to have melanoma of the head and neck (14% vs. 7%, p<0.001) and less likely to have ulceration (1% vs 10%, p<0.001), regression (26% vs. 45%, p<0.001), and a Clark level IV-V (6% vs. 26%, p<0.001). They also had lower mitotic rates. With the matching we have created two more homogeneous groups (at least with respect to these variables) at the expense of having to give up a significant number of patients.  There were no significant differences between the groups after matching.

in the abstract, “thin” melanoma is mentioned and not described. The definition of this being less than 1.2 mm Breslow thickness should be stated in abstract. 

Answer: We agree the reviewer. It has  been added. We have modified the manuscript as follows: The aim of the study is to determine the role of SLNB in the survival of thin melanomas (≤1 mm).

Reviewer 2 Report

Thank you for that excellent multicenter study. My remarks are as below:

  1. could you please provide other histological types' descriptions? it's 588 cases, and you decided to extract only SMM and nodular type.
  2. you decided to include only thin melanomas - could you explain how it is possible that you have Clark IV and V? 
  3.  could you present some of the multivariate analysis with graphs?
  4. could you deeper discuss and characterize the "high-risk thin melanoma group"?

Author Response

Dear editor,

We would like to thank the reviewers for his comments which have contributed to the improvement in our manuscript

Reviewer 2

Thank you for that excellent multicenter study. My remarks are as below:

Could you please provide other histological types' descriptions? it's 588 cases, and you decided to extract only SMM and nodular type.

Answer: For the convergence of the models it was mandatory to reunified the rest of histologic subgroups (lentigo maligna, acral lentiginous melanoma, …) into a single simple group (others). Keeping all histological subtypes individually would have made it very difficult to obtain so many patients after propensity score matching.

 You decided to include only thin melanomas - could you explain how it is possible that you have Clark IV and V?  

Answer: The observation of Clark IV and V in thin melanomas, although uncommon,  is referenced in approximately 10-15% of cases. (for example in: Owen SA et al. Identification of higher risk thin melanomas should be based on Breslow depth not Clark level IV. Cancer. 2001; 91:983-91). In fact, the presence of a Clark level IV or V had been a classical criteria to indicate SLNB in thin melanoma.

Could you present some of the multivariate analysis with graphs?

Answer: Due to the nature  of the study and the limitations of the journal’s author guidelines we decided  to show  results with tables. Drawing  a graph of multivariate analysis would be an alternative and interesting method to present our results. However we believe that it would provide the same information and probably the reader is more used to reading this type of information in the form of a table.

Could you deeper discuss and characterize the "high-risk thin melanoma group"?

Answer: thank you for this suggestion. It has been added a comment in discussion concerning ulceration, thickness, and age in relation with survival. We have modified the manuscript as follows: These data are congruent with the AJCC as ulceration is established as a variable that increases the staging according to a certain thickness while melanoma thicknesses close to 1 mm are already considered another stage[33]. It remains to be seen whether advanced age may contribute in the future to define these melanomas with a worse prognosis.

Round 2

Reviewer 1 Report

I accept the modifications made by the research team..

Author Response

We appreciate the answer of the reviewer